# Effect of pragmatic versus explanatory interventions on medication adherence in people with cardiometabolic conditions: a systematic review and meta-analysis

Claire Fitzpatrick,[1,2] Clare Gillies,[1,2] Samuel Seidu,[1,2] Debasish Kar,[3] Ekaterini Ioannidou,[1,2] Melanie J Davies,[1,2] Prashanth Patel,[4,5] Pankaj Gupta,[4,5] Kamlesh Khunti [1,2,6,7]

CF and CG contributed equally.

For numbered affiliations see end of article.

**Correspondence to**
Professor Kamlesh Khunti;
kk22@le.ac.uk

## ABSTRACT

**Objective** To synthesise findings from randomised controlled trials (RCTs) of interventions aimed at increasing medication adherence in individuals with type 2 diabetes (T2DM) and/or cardiovascular disease (CVD). And, in a novel approach, to compare the intervention effect of studies which were categorised as being more pragmatic or more explanatory using the Pragmatic-Explanatory Continuum Indicator Summary-2 (PRECIS-2) tool, to identify whether study design affects outcomes. As explanatory trials are typically held under controlled conditions, findings from such trials may not be relatable to real-world clinical practice. In comparison, pragmatic trials are designed to replicate real-world conditions and therefore findings are more likely to represent those found if the intervention were to be implemented in routine care.

**Design** Systematic review and meta-analysis.

**Data sources** Ovid Medline, Ovid Embase, Web of Science and CINAHL from 1 January 2013 to 31 December 2018.

**Eligibility criteria for selecting studies** RCTs lasting ≥3 months (90 days), involving ≥200 patients in the analysis, with either established CVD and/or T2DM and which measured medication adherence. From 4403 citations, 103 proceeded to full text review. Studies published in any language other than English and conference abstracts were excluded.

**Main outcome measure** Change in medication adherence.

**Results** Of 4403 records identified, 34 studies were considered eligible, of which 28, including 30 861 participants, contained comparable outcome data for inclusion in the meta-analysis. Overall interventions were associated with an increase in medication adherence (OR 1.57 (95% CI: 1.33 to 1.84), p<0.001; standardised mean difference 0.24 (95% CI: −0.10 to 0.59) p=0.101). The effectiveness of interventions did not differ significantly between studies considered pragmatic versus explanatory (p=0.598), but did differ by intervention type, with studies that included a multifaceted rather than a single-faceted intervention having a more significant effect (p=0.010). The analysis used random effect models and used the revised Cochrane Risk of Bias Tool to assess study quality.

## Strengths and limitations of this study

► In a novel approach, this systematic review compared whether study design (pragmatic vs explanatory as defined by the Pragmatic-Explanatory Continuum Indicator Summary-2 (PRECIS-2) tool) had an impact on intervention effect.
► The study selection was undertaken independently by two researchers to ensure that all relevant studies were included as well as preventing the risk of individual biases on study selection.
► The impact of study heterogeneity was explored using subgroup analyses.
► This review provides a contemporary update to the 2014 Cochrane Review on medication adherence.
► A potential limitation of this systematic review and meta-analysis is that a small number of studies had to be excluded as they did not categorise diabetes type.

**Conclusions** In this meta-analysis, interventions were associated with a significant increase in medication adherence. Overall multifaceted interventions which included an element of education alongside regular patient contact or follow-up showed the most promise. Effectiveness of interventions between pragmatic and explanatory trials was comparable, suggesting that findings can be transferred from idealised to real-word conditions.

**PROSPERO registration number** CRD42017059460.

## INTRODUCTION

Prevalence of type 2 diabetes (T2DM) and cardiovascular diseases (CVDs) are increasing rapidly.[1] They have been identified as two of the most common cardiometabolic morbidities associated with multimorbidity[1] and are two of the leading causes of death worldwide.[2] Most people diagnosed with these conditions will likely have multimorbidity[1] (coexistence

of two or more chronic conditions), particularly those aged 65–84 years where prevalence is estimated at 65%.[3]

Management of multimorbidity is complex, typically relying on the coprescription of multiple drugs, which is strongly associated with medication non-adherence.[4] The WHO estimates that 50% of individuals receiving chronic treatment are non-adherent (taking less than 80%) to prescribed medications.[5] Medication non-adherence is considered the biggest cause of suboptimal clinical outcomes, accounting for approximately 57% of avoidable costs in relation to medication use.[5]

Despite the rise in cardiometabolic multimorbidity,[6–8] interventions to increase medication adherence within this population are sparse. As such we decided to focus this review on T2DM and CVD as the most prevalent cluster cardiometabolic morbidities. Treatment non-adherence within these populations is well recognised and has become the focus of considerable research. While randomised controlled trials (RCTs) are widely accepted as a rigorous way of exploring the impact of interventions on specific health behavioural change outcomes, it has been identified that numerous components within a trial design can lead to biased interpretations of intervention effects.[9] One such bias may be the controlled nature of these trials and the impact they impose on the cooperation of their participants and may prelude an action which does not necessarily represent what may occur in routine clinical practice.[10] In pragmatic trials, the intervention is less strict and mimics usual practice as much as possible, thus lessening the unexpected reactions from the patients which lead to the biases.[11] As RCTs are generally expensive to conduct, it is important that their findings show real-world intervention effectiveness that is relevant to routine clinical practice. This study therefore aimed to not only identify interventions to increase medication adherence but also compare the effectiveness of interventions categorised as explanatory (undertaken in idealised settings) or pragmatic (undertaken in real-world settings). Described by Schwartz and Lellouch,[12] explanatory trial are those which confirm a physiological or clinical hypothesis; in contrast, pragmatic trails are those which inform clinical or policy decisions by evidencing the effect that adoption would have on routine care. As treatment effects of explanatory trials may be larger than those observed in pragmatic trials, traditional meta-analytic approaches may not account for this heterogeneity resulting in biased estimated treatment effects. While a handful of reviews in different research areas have now been published which retrospectively applied PRECIS-2 (Pragmatic-Explanatory Continuum Indicator Summary-2) to see if comparing pragmatic with explanatory trials altered review findings,[13 14] to the authors knowledge, this is the first review to incorporate PRECIS-2 scoring into the initial review process. While retrospective application provides an interesting comparison, whereby the overall intervention effects can be compared with explanatory and/or pragmatic intervention effects, pre- or post-use of PRECIS-2, it could lead to an initial misinterpretation of findings. For example, a review containing a high number of explanatory trials may be much less applicable to routine clinical practice than one containing a greater number of pragmatic trials. Identification and comparison during the trial design stage would allow researchers to more easily identify how applicable findings would be to real-world clinical practice, rather than making an assumption based on a generalised outcome. In addition, this approach removes any risk of bias as the analysts are unaware of the results of the study. Using PRECIS-2 this review scored interventions from very explanatory to very pragmatic to explore whether the differences in characteristics between the study designs of these trials influenced intervention effectiveness estimates in a meta-analysis.

## METHODS

This systematic review has been registered on PROSPERO and was conducted in accordance with PRISMA (Preferred Reporting Items for Systematic Reviews and Meta-Analyses) guidelines.[15]

### Data sources and searches

An extensive scoping search identified the great breath of research published within this topic area. It also identified a 2014 Cochrane Review on medication adherence which included studies published up to 11 January 2013. Due to factors relating to the focus of the review, changes in prescribed medication and the increasing use of mobile technology interventions in recent adherence research, we decided that this review would provide a contemporary update and therefore we refined our literature search to include studies published between 1 January 2013 to 31 December 2018. We searched Ovid Medline 1946 (Epub ahead of print, in process and other non-indexed citations), Ovid Embase 1974, Web of Science 1970 and Cumulative Index of Nursing and Allied Health Literature (EBSCO CINAHL-plus with full text). We searched "type 2 diabetes", "cardiovascular disease", "medication adherence" and "randomised control trial" using a combination of medical subject headings, keywords and synonyms with both English and American spellings. An example search strategy can be found in online supplementary etable 1.

### Screening

Two investigators (CF and EI) independently screened all titles, abstracts and full text articles. Discrepancies were resolved through discussion with a third researcher (CG). Studies were assessed against five eligibility criteria to determine first, whether the study was an RCT; second, that patients were identified as having established CVD and/or T2DM; third, the trial measured medication adherence; fourth, study duration was ≥3 months (90 days) and fifth, the study included >200 people in the analysis. Studies published in any language other than English and conference abstracts were excluded. Additional reasons

for exclusion were where articles failed to specify diabetes type or where results were combined for individuals at risk of and with established CVD.

## Data extraction

Data extraction was performed by CF and checked by CG. Where available the following data were recorded: study characteristics (authors, year, country, duration, sample size), participant characteristics (age, gender, disease), methods of assessment (self-report, pharmacy records, pill count), intervention type (online supplementary etable 2) and outcome measures of medication adherence. For trials reporting multiple follow-up, the final follow-up corresponding to study duration was used.

## Risk of bias

Risk of bias assessment was undertaken using the revised Cochrane Risk of Bias Tool for RCTs.[16 17]

## Pragmatic-Explanatory Continuum Indicator Summary-2

Using Loudon *et al* (2015) for guidance,[18] PRECIS-2 scoring to explore how pragmatic/explanatory different components of the included study designs was undertaken independently by CF and CG. Each of the nine domains was given a score from 1 (very explanatory 'ideal study conditions') to 5 (very pragmatic 'usual care conditions'). Results were compared and a consensus score was reached.

Some studies did not clearly explain all components making scoring difficult. To determine whether the treatment of unclear data affected the overall score, we conducted two classification processes. For the primary method we inputted a score of 3 and then calculated an average score by adding up the scores and dividing by 9. In the second, a score was not given for the missing domains and then an average was calculated based on available domain scores.

## Data synthesis

Study data were reported as means and medians for continuous data and as proportions for categorical data. Twenty-two studies defined a cut point to determine adherence. Twenty used a binary outcome defining individuals as adherent or non-adherent; however, two categorised individuals based on a prespecified level of low, medium and high. Of those reporting levels of low, medium or high adherence, as has been done in previous studies,[19 20] we combined medium and high levels and separated low level to form a binary outcome of adherence or non-adherence, respectively, therefore enabling OR of adherence to be calculated. In one study (Boyne *et al*, 2014), all participants in the intervention arm were adherent at follow-up, so a continuity correction of 0.5 was added to allow an OR to be calculated. Where an OR for medication adherence was reported in study results, this was used for the meta-analyses, rather than calculating an estimated OR from the raw numbers. Where adjusted ORs were reported, the OR adjusted for the most covariates was used in the meta-analyses.

A further six studies, plus one which was included in the previous analysis, reported adherence using a continuous scale which provided an overall group indication of adherence. For these studies, change in adherence for each study arm was calculated. As scores differed, we calculated standardised mean differences (SMDs) to combine study estimates in a meta-analysis. The SMD was calculated as the mean change in medication adherence in the intervention group minus mean change in the control group, divided by the pooled SD, using Cohen's method.[21]

For both meta-analysis models, funnel plots and Egger's test were carried out to assess for publication bias, and random effects meta-analyses were fitted to account for heterogeneity in study design. We assessed heterogeneity between studies by calculating the Higgins $I^2$ statistic, with an $I^2$ statistic >75% considered high heterogeneity.[22] In one study (Boyne *et al*, 2014), all participants in the intervention arm were adherent at follow-up, so a continuity correction of 0.5 was added to allow an OR to be calculated.

We fitted meta-regression models assessing; study length, whether adherence was a primary or secondary outcome, mean age, percent male, disease (T2DM or CVD), and PRECIS-2 score, to explore the impact of study heterogeneity on the intervention effect. Three subgroup analyses were also carried out, whereby fitting a meta-regression model to compare the statistical difference between groups, separate meta-analyses were also run for each subgroup to enable the pooled estimate for each subgroup to be calculated, and hence a more explicit comparison to be made. The first compared intervention effects of studies that included a self-reported measure of adherence, to those with an objective adherence measure. The second compared pragmatic with explanatory studies (pragmatic studies were those with an average PRECIS-2 score >3 and explanatory a score of ≤3) and the third compared outcomes of interventions which were identified as multifaceted against those with a singular intervention component.

Where studies had obtained both pharmacy and self-report data, pharmacy data were used in preference in the meta-analysis. In the case of a few studies which did not report overall level of adherence and instead listed individual drug adherence, we made the pragmatic decision to monitor the adherence effect of the first drug listed by the author.

All analyses were performed using Stata 15 with the METAN, METAREG and METABIAS commands. Results are reported with 95% CIs with a p value <0.05 considered statistically significant. For the few studies not included within the meta-analysis, descriptive synthesis of intervention effects is discussed.

## RESULTS
### Study selection

Following deduplication we screened 4403 titles and abstracts which yielded 103 potentially relevant studies. Thirty-three met the inclusion criteria with an additional study[23] identified from reference list and forward citation

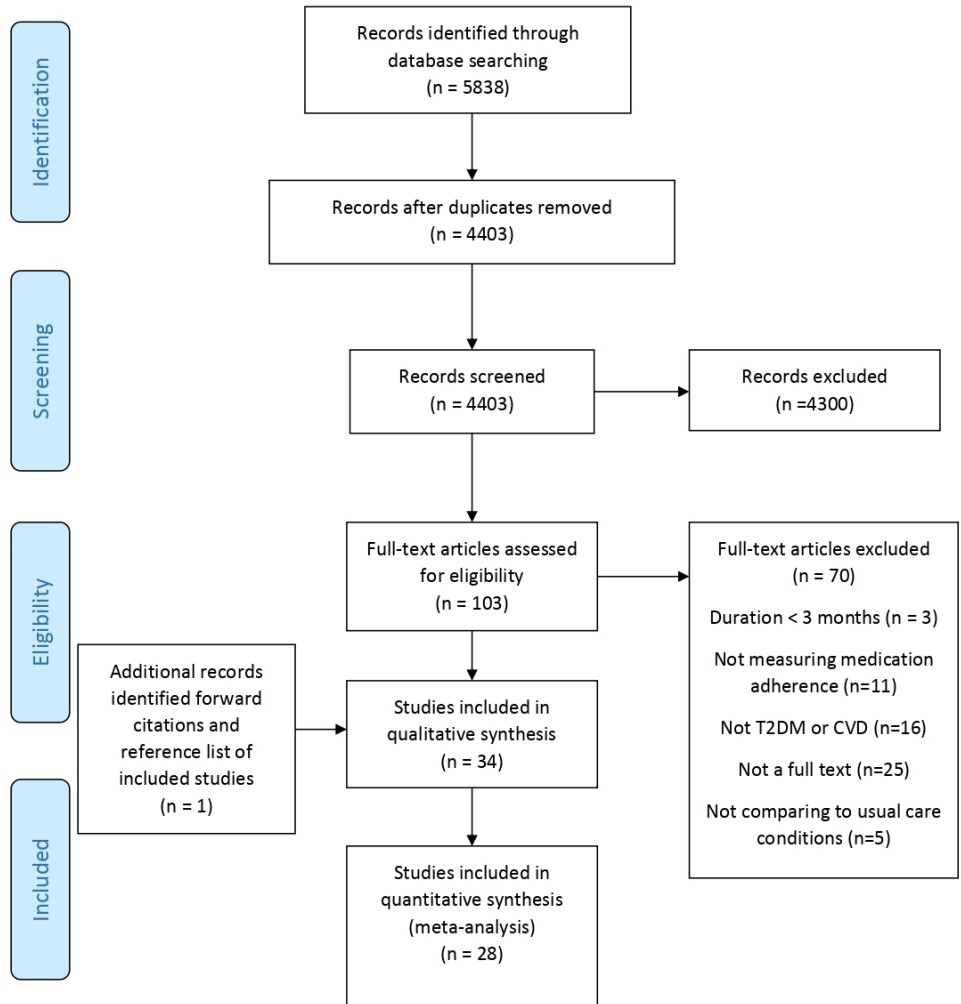

**Figure 1** PRISMA (Preferred Reporting Items for Systematic Reviews and Meta-Analyses) flow chart of included studies. CVD, cardiovascular disease; T2DM, type 2 diabetes mellitus.

screening (figure 1).[23] Of the 34 studies, 28 including 30 861 participants contained comparable outcome data for inclusion in the meta-analysis.

## Characteristics of included studies

An overview of the characteristics of the included studies is available in table 1. A more detailed key characteristics of included studies is available in online supplementary etable 3. Studies were conducted in 17 countries, with one study[24] conducted across 4 countries. Twenty-three trials were conducted in high-income countries,[23 25–46] seven in upper middle,[47–53] two in lower middle,[54 55] one in a low-income country and one across both upper middle and high income economies.[56] The proportion of males ranged from 27.9%[31] to 94.0%.[23] Four trials did not report age[24 26 52 56] and/or gender.[24 26 56]

Medical conditions varied across studies; 12 studies were in participants with T2DM,[25 28–31 34 37 41 48 53 54 56] 21 in participants with CVD[23 24 26 27 32 33 35 36 38–40 42–44 46 47 49–52 55] and 1 in people with both.[45] Follow-up ranged from 3 to 36 months.

Medication adherence was assessed as a primary outcome in 19 studies.[24 25 27–29 33 35 36 39 42 44 45 48 50–53 55 56] Methods of measuring medication adherence varied between studies. Adherence was primarily assessed through self-report (n=23),[23–32 34 38 40 42 44 47–50 53–56] followed by pharmacy data (n=10),[28 35 36 41–43 45 46 50 53] pill count (n=2)[24 33] and electronic pill bottle opening (n=1).[39] Three studies did not report the method of medication adherence assessment used.[37 51 52] Of those measuring adherence through self-report, the Morisky-Green Questionnaire[57] and the 8-item Morisky Medication Adherence Scale (MMAS-8)[58] (or translated versions) were the most frequently used. Four studies measured adherence using two different methods: self-report and either pharmacy or pill count data.[24 28 50 53] Two of the studies combined the data to provide an overall level or adherence,[24 28] whereas the other two reported adherence outcomes separately.[50 53] Of the two studies reporting separate outcomes, both reported consistent findings and statistically significant improvements in adherence whether self-report or pharmacy data were used.

Interventions varied greatly between studies. We categorised and defined interventions (online supplementary etable 2) based on their most prominent feature into one

**Table 1** Overview of characteristics of included studies

| First author, year | Sample size (analysed) Duration (data collection points) | Condition | Key intervention detail | Medication adherence measure |
|---|---|---|---|---|
| Al-Haj Mohd, 2016[25] | 446 (446) 6 months (0 and 6 months) | T2DM | Multifaceted intervention strategy | Self-report (MMAS-8) |
| Barker-Collo, 2015[26] | 386 (326) 12 months (0, 3, 6, 9 and 12 months) | Stroke/TIA | Behavioural/educational | Self-report with crosscheck of pharmacy data |
| Boyne, 2014[27] | 382 (382) 12 months (0, 3, 6 and 12 months) | Heart failure | Telemonitoring/telemedicine | Self-report (European Heart Failure Self-care Behaviour Scale) |
| Buhse, 2017[28] | 279 (279) 6 months (0 and 6 months) | T2DM | Behavioural/educational | Pharmacy data Self-report (interview) |
| Caetano, 2018[29] | 709 (702) 6 months (0 and 6 months) | T2DM | Behavioural/educational | MAT scale |
| Cao, 2017[47] | 236 (236) 90 days (0, 30 and 90 days) | Coronary heart disease | Collaborative care | Self-report (MMAS-8) |
| Carrasquillo, 2017[30] | 300 (215) 12 months (0 and 12 months) | T2DM | Behavioural/educational | Self-report (MMAS-8) |
| Castellano, 2014[24] | 695 (594) 9 months (0, 1, 4 and 9 months) | Myocardial infarction | Simplification of drug regimen | Pill count Self-report (Morisky-Green-Levine Adherence Scale) |
| Chung, 2014[48] | 241 (241) 12 months (0, 4, 8 and 12 months) | T2DM | Multifaceted intervention strategy | Self-report—MMAS-8 (Revised Malaysian version) |
| Crowley, 2013[31] | 359 (329) 12 months (0, 3, 6, 9 and 12 months) | T2DM | Multifaceted intervention strategy | Self-report (Morisky-Green-Levine Adherence Scale) |
| Du, 2016[32] | 979 (964) 36 months (0 and 36 months) | Percutaneous coronary intervention | Multifaceted intervention strategy | Self-report (Morisky-Green-Levine Adherence Scale) |
| El-Touky, 2017[33] | 321 (276) 12 months (0, 1, 3, 6, 9 and 12 months) | Acute coronary syndrome | Behavioural/educational intervention | Pill count |
| Graumlich, 2016[34] | 674 (674) 12 months (0, immediately postintervention, 3 and 6 months) | T2DM | Medication monitoring table | Self-report (PMAQ) |
| Hedegaard, 2015[35] | 211 (203) 12 months (0, 3, 6, 9 and 12 months) | Stroke/TIA | Multifaceted intervention strategy | Pharmacy data—MPR |
| Ho, 2014[36] | 253 (241) 12 months (0 and 12 months) | Acute coronary syndrome | Multifaceted intervention strategy | Pharmacy refill data |
| Jeong, 2018[37] | 338 (338) 24 weeks (0 and 24 weeks) | T2DM | Telemonitoring/telemedicine | Does not report |
| Jia, 2017[49] | 669 (669) 36 months (0, 1, 3, 6, 12 and 36 months) | Percutaneous coronary intervention | Intensified patient care | Self-report (Morisky-Green-Levine Adherence Scale) |
| Kronish, 2014[38] | 600 (600) 6 months (0 and 6 months) | Stroke/TIA | Behavioural/educational intervention | Self-report (MMAS-8) |
| Lin, 2017[50] | 288 (288) 18 months (0, 6, 12 and 18 months) | Coronary artery bypass grafting | Multifaceted intervention strategy | Self-report—MARS (5 item) |
| Marin, 2015[56] | 467 (459) 12 months (0 and 12 months) | T2DM | Personalised medication management | Self-report (Morisky-Green-Levine Adherence Scale) |
| Marquez-Contreras, 2018[39] | 726 (625) 18 months (0, 6 and 12 months) | Atrial fibrillation | Multifaceted intervention strategy | MEMs |

**Table 1**   Continued

| First author, year | Sample size (analysed) Duration (data collection points) | Condition | Key intervention detail | Medication adherence measure |
|---|---|---|---|---|
| Meng, 2014[23] | 471 (425) 12 months (admission, discharge, 6 and 12 months) | Coronary heart disease | Behavioural/educational intervention | Self-report—MARS-D |
| Meng, 2016[40] | 513 (449) 12 months (admission, discharge, 6 and 12 months) | Heart failure | Behavioural/educational intervention | Self-report—MARS-D |
| Peng, 2014[51] | 3821 (3330) 12 months (discharge, 6, 9 and 12 months) | Stroke/TIA | Behavioural/educational intervention | Does not report |
| Pladevall, 2015[41] | 1692 (1512) 18 months (0, 6, 12 and 18 months) | T2DM | Behavioural/educational intervention | Pharmacy data—PDC |
| Rinfret, 2013[42] | 300 (300) 12 months (0 and 12 months) | Percutaneous coronary intervention | Intensified patient care | Pharmacy refill data |
| Samtia, 2013[54] | 348 (348) 5 months (0 and 5 months) | T2DM | Behavioural/educational | Self-report |
| Schou, 2014[43] | 921 (920) 13–72 months (every 1–3 months) | Heart failure | Multifaceted intervention strategy | Pharmacy data—PDC |
| Schwalm, 2015[44] | 852 (852) 12 months (0, 3 and 12 months) | Myocardial infarction | Behavioural/educational | Self-report (Morisky-Green-Levine Adherence Scale) |
| Su, 2017[52] | 1275 (1187) 12 months (0 and 12 months) | Stroke/TIA | Multifaceted intervention strategy | Does not report |
| Vollmer, 2014[45] | 21 752 (21 752) 12 months (0 and 12 months) | T2DM and/or CVD | Telemonitoring/telemedicine | Pharmacy data—PDC |
| Volpp, 2017[46] | 1509 (1503) 12 months (0 and 12 months) | Myocardial infarction | Multifaceted intervention strategy | Pharmacy data—PDC |
| Xavier, 2016[55] | 805 (750) 12 months (0 and 12 months) | Acute coronary syndrome | Multifaceted intervention strategy | Pharmacy data—composite medical adherence score >80% |
| Xin, 2015[53] | 240 (227) 12 months (0 and 12 months) | T2DM | Behavioural/educational | Prescription refill claims Self-report (Morisky-Green-Levine Adherence Scale) |

CVD, cardiovascular disease; MARS-D, Medication Adherence Report Scale (German version); MAT, Measure of Adherence to Treatments; MEMs, Medical Event Monitoring Systems; MMAS-8, Morisky Medication Adherence Scale (8-item); 0 months, baseline; MPR, medication possession ratio; PDC, proportion of days covered; PMAQ, Patient Medication Adherence Questionnaire; T2DM, type 2 diabetes mellitus; TIA, transient ischaemic attack.

of the following seven groups: (1) behavioural/educational,[23 26 28–30 33 34 38 40 41 44 51 53 54] (2) intensified patient care,[42 49] (3) collaborative care,[47] (4) simplification of drug regimen,[24] (5) personalised drug dispensing,[56] (6) telemonitoring/telemedicine[27 37 45] and (7) multifaceted intervention strategy.[25 31 32 35 36 39 43 46 48 50 52 55]

### PRECIS-2 scoring

Details of PRECIS-2 scoring for each of the included studies can be found in table 2. For the purpose of this study it was considered that each of the nine domains on the PRECIS-2 wheel had equal weighting. For visual clarity, PRECIS-2 results are presented as a shaded graph (table 2) where darker shades represent more pragmatic components. We also inputted scores on the PRECIS-2 wheel (online supplementary efigures 1–3).

Of the 34 studies, 20 studies were identified as being more pragmatic, with an average score >3 (range 3.11–4.11).[23 24 26 27 29 32 35–37 40–49 52] Of these, most (n=18) received an average score between 3 and 4, demonstrating a slightly more pragmatic intention on the PRECIS-2 continuum. Two of the studies scored ≥4,[29 45] demonstrating a greater degree of pragmatism. The majority of these studies scored poorly (score of 1) in relation to pragmatism of the primary outcome as, based on the guidance by Loudon et al,[18] medication adherence was not considered of obvious importance from the patients' perspective and was typically assessed by methods not

**Table 2**  Shaded graph to show PRECIS-2 scoring of included studies

| First author | Year | Eligibility | Recruitment | Setting | Organisation | Flexibility: delivery | Flexibility: adherence | Follow-up | Primary outcome | Primary analysis | Total score | Average score | Amended total score | Amended average score |
|---|---|---|---|---|---|---|---|---|---|---|---|---|---|---|
| Al-Haj Mohd [25] | 2016 | 4 | 4 | 5 | 1 | 2 | 3 | 2 | 1 | 5 | 27 | 3 | 24 | 3 |
| Barker-Collo [26] | 2015 | 5 | 5 | 5 | 1 | 1 | 3 | 1 | 5 | 5 | 31 | 3.44 | 28 | 3.5 |
| Boyne [27] | 2014 | 5 | 5 | 5 | 2 | 1 | 3 | 1 | 2 | 5 | 29 | 3.22 | 26 | 3.25 |
| Buhse [28] | 2017 | 1 | 4 | 5 | 1 | 2 | 3 | 1 | 1 | 5 | 23 | 2.56 | 20 | 2.5 |
| Caetano [29] | 2018 | 4 | 5 | 5 | 4 | 5 | 5 | 3 | 1 | 5 | 37 | 4.11 | 34 | 4.25 |
| Cao [47] | 2017 | 2 | 4 | 5 | 1 | 2 | 4 | 2 | 5 | 5 | 30 | 3.33 | 30 | 3.33 |
| Carrasquillo [30] | 2017 | 1 | 1 | 4 | 1 | 2 | 1 | 1 | 2 | 4 | 17 | 1.89 | 17 | 1.89 |
| Castellano [24] | 2014 | 4 | 5 | 5 | 5 | 5 | 2 | 2 | 1 | 5 | 34 | 3.78 | 34 | 3.78 |
| Chung [48] | 2014 | 2 | 5 | 5 | 4 | 1 | 5 | 2 | 1 | 5 | 30 | 3.33 | 30 | 3.33 |
| Crowley [31] | 2013 | 1 | 1 | 5 | 2 | 2 | 3 | 2 | 2 | 4 | 22 | 2.44 | 19 | 2.38 |
| Du [32] | 2016 | 5 | 4 | 5 | 2 | 2 | 3 | 2 | 5 | 5 | 33 | 3.67 | 30 | 3.75 |
| El-Touky [33] | 2017 | 4 | 5 | 5 | 4 | 3 | 1 | 2 | 1 | 1 | 26 | 2.89 | 23 | 2.88 |
| Graumlich [34] | 2016 | 1 | 2 | 5 | 1 | 1 | 4 | 2 | 2 | 5 | 23 | 2.56 | 23 | 2.56 |
| Hedegaard [35] | 2015 | 4 | 4 | 5 | 2 | 2 | 3 | 4 | 1 | 5 | 30 | 3.33 | 27 | 3.38 |
| Ho [36] | 2014 | 4 | 4 | 5 | 2 | 1 | 3 | 5 | 1 | 5 | 30 | 3.33 | 27 | 3.38 |
| Jeong [37] | 2018 | 4 | 5 | 5 | 1 | 4 | 3 | 2 | 2 | 4 | 30 | 3.33 | 27 | 3.38 |
| Jia [49] | 2017 | 3 | 4 | 5 | 2 | 4 | 3 | 4 | 1 | 4 | 30 | 3.33 | 27 | 3.38 |
| Kronish [38] | 2014 | 4 | 1 | 4 | 1 | 2 | 3 | 1 | 2 | 5 | 23 | 2.56 | 20 | 2.5 |
| Lin [50] | 2017 | 1 | 2 | 4 | 1 | 2 | 3 | 1 | 1 | 5 | 20 | 2.22 | 17 | 2.13 |
| Marin [56] | 2015 | 1 | 2 | 5 | 4 | 4 | 3 | 2 | 1 | 4 | 26 | 2.89 | 23 | 2.88 |
| Marquez-Contreras [39] | 2018 | 2 | 1 | 5 | 2 | 2 | 1 | 3 | 2 | 2 | 19 | 2.11 | 19 | 2.11 |
| Meng [23] | 2014 | 4 | 5 | 5 | 2 | 4 | 3 | 2 | 2 | 4 | 31 | 3.44 | 28 | 3.5 |
| Meng [40] | 2016 | 2 | 5 | 4 | 2 | 4 | 3 | 2 | 4 | 4 | 30 | 3.33 | 27 | 3.38 |
| Peng [51] | 2014 | 3 | 5 | 5 | 2 | 1 | 3 | 3 | 1 | 4 | 27 | 3 | 21 | 3 |
| Pladevall [41] | 2015 | 2 | 2 | 5 | 1 | 5 | 3 | 4 | 2 | 5 | 29 | 3.22 | 26 | 3.25 |
| Rinfret [42] | 2013 | 4 | 4 | 5 | 2 | 4 | 3 | 4 | 1 | 5 | 32 | 3.56 | 29 | 3.63 |
| Samtia [54] | 2013 | 2 | 4 | 4 | 4 | 3 | 3 | 1 | 2 | 4 | 27 | 3 | 21 | 3 |
| Schou [43] | 2014 | 1 | 5 | 5 | 4 | 4 | 4 | 4 | 1 | 4 | 32 | 3.56 | 32 | 3.56 |
| Schwalm [44] | 2015 | 5 | 5 | 5 | 4 | 4 | 3 | 2 | 1 | 4 | 33 | 3.67 | 30 | 3.75 |
| Su [52] | 2017 | 4 | 5 | 5 | 2 | 2 | 3 | 2 | 1 | 4 | 28 | 3.11 | 25 | 3.13 |
| Vollmer [45] | 2014 | 4 | 5 | 5 | 4 | 4 | 3 | 5 | 1 | 5 | 36 | 4 | 33 | 4.13 |
| Volpp [46] | 2017 | 2 | 1 | 5 | 1 | 4 | 2 | 5 | 5 | 5 | 30 | 3.33 | 30 | 3.33 |
| Xavier [55] | 2016 | 4 | 5 | 5 | 1 | 2 | 2 | 1 | 1 | 5 | 26 | 2.89 | 26 | 2.89 |
| Xin [53] | 2015 | 2 | 3 | 3 | 2 | 4 | 3 | 2 | 1 | 4 | 24 | 2.67 | 15 | 2.5 |

PRECIS-2, Pragmatic-Explanatory Continuum Indicator Summary-2.

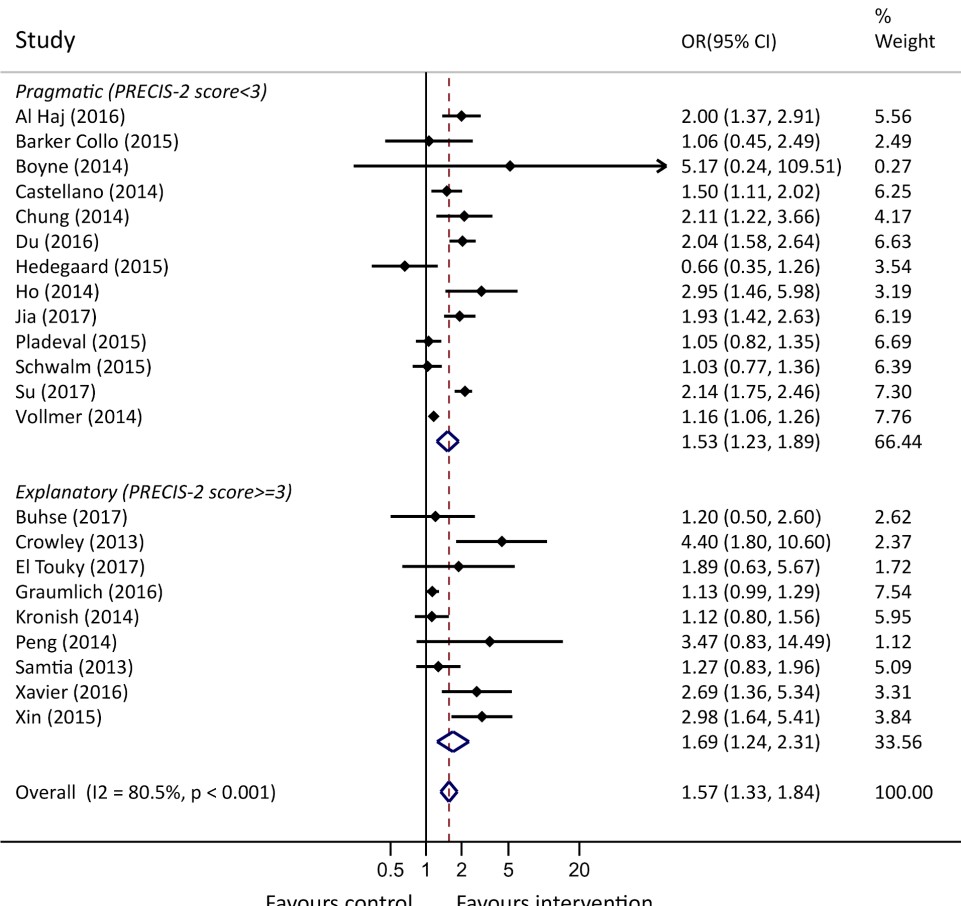

**Figure 2** Forest plot of pooled ORs for medication adherence, stratified by Pragmatic-ExplanatoryContinuum Indicator Summary-2 (PRECIS-2) score.

used routinely in primary care. Eleven studies were identified as being more explanatory receiving an average score of <3 (range 1.89–2.89).[28 30 31 33 34 38 39 50 53 55 56] Of these, most (n=10) received an average score between 2 and 3 demonstrating that they favour a more explanatory intention on the continuum. Finally, three studies were given a score of 3[25 51 54] suggesting that these were equally pragmatic and explanatory.

The domains within which trials scored a more pragmatic rating were recruitment, setting of the intervention and the primary analysis which received average scores of 3.7, 4.8 and 4.4, respectively. All other domains appeared more explanatory with average scores of 2.2, 2.8, 2.9, 2.4 and 1.8 for the domains of organisation, flexible delivery, flexible adherence, follow-up and primary outcome, respectively. The eligibility domain was the only one to receive a score of 3.

When results were compared based on the way missing data were scored, no changes to any studies overall categorisation was observed.

## META-ANALYSIS
### Effect of interventions
Data from 22 studies were pooled in a meta-analysis (figure 2). Irrespective of intervention type or duration,

interventions significantly improved medication adherence when compared with control (OR 1.57 (95% CI: 1.33 to 1.84), p<0.001). For the seven studies where adherence was reported as a continuous variable, the pooled SMD was estimated as 0.24 ((95% CI: −0.10 to 0.59), p=0.101) (online supplementary efigure 4), in favour of the intervention group, but was not statistically significant.

Both meta-analyses showed statistically significant heterogeneity between studies (p<0.001) with high $I^2$ values of 80.5% and 95.4% for the meta-analyses of the ORs and SMDs, respectively, where the $I^2$ value represents the percentage of variability due to heterogeneity rather than sampling error.

We investigated the heterogeneity by conducting meta-regression analyses (online supplementary etable 4). No statistically significant association was found between the intervention effects and study length, mean study age, male per cent, disease (CVD or T2DM), whether it was a primary or secondary outcome or PRECIS-2 score. Subgroup analyses identified no differences in the intervention effect between self-reported and objective measures of adherence, and between PRECIS-2 score (online supplementary etable 5). Subgroup analyses comparing multifaceted versus singular-faceted interventions showed that multifaceted interventions led to

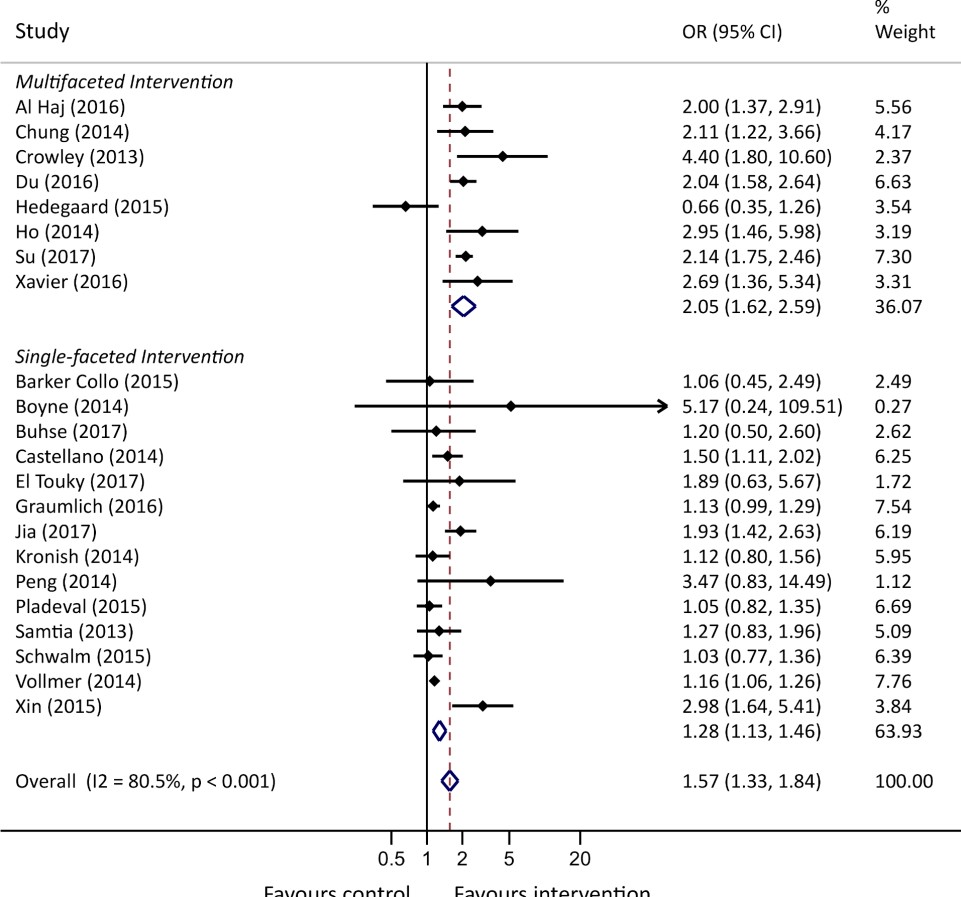

**Figure 3** Forest plot of ORs for adherence, stratified by complexity of the intervention.

a statistically significant improvement in odds of adherence (p=0.010) (figure 3; online supplementary figure S3, etable 5). The funnel plot and Egger's test (p=0.184) showed that no indication of publication bias was present for the meta-analysis of SMDs (online supplementary efigure 5). For the meta-analysis of ORs, Egger's test for publication bias was statistically significant (p=0.041), with the funnel plot asymmetry indicating a slightly uneven presence of small studies showing a favourable intervention effect (online supplementary efigure 6).

### Descriptive synthesis
Six studies did not contain comparable data for inclusion in the meta-analysis.[37 39 42 43 46 56] Of those, four studies showed a positive intervention effect on rates of medication adherence,[37 39 42 56] while two showed no statistically significant intervention effects.[43 46]

### Assessment of risk of bias
Table 3 displays trial specific risk of bias assessment. Few studies were deemed to be of fair quality with low risk of bias and none were considered to be free of bias. Twenty-seven trials provided information about adequate sequence generation and 17 regarding allocation concealment. Incomplete outcome data was a concern in four trials and selective reporting was primarily deemed unclear (n=19) due to the number of trials without

a published protocol. Blinding of participants and personnel was the main domain scoring poorly for risk of bias; however, the nature of these types of trials means it is often impractical to do so.

### DISCUSSION
Medication adherence is a key component of chronic disease care yet many patients fail to follow prescribing guidelines.[5] This review identified 34 trials that evaluated the impact of interventions aimed at improving medication adherence in individuals with either T2DM and/ or established CVD. Overall interventions significantly increased medication adherence.

To our knowledge this is the first review to systematically compare and synthesise medication adherence outcomes based on the PRECIS-2 classification of trials. Our review showed that interventions improve medication adherence and the PRECIS-2 classification did not appear to affect the outcome when compared in subgroup and meta-regression analyses. This finding differs from previous research where explanatory trials have reported significantly larger effect sizes than pragmatic trials.[13 14] This suggests that findings from these interventions are representation of real-world clinical practice.

**Table 3** Revised Cochrane risk of bias of included studies

| First author, year | Adequate random sequence generation | Allocation concealment | Blinding of participants | Blinding of personnel | Blinding of outcome assessors | Incomplete outcome data | Selective reporting | Other |
|---|---|---|---|---|---|---|---|---|
| Al-Haj Mohd[25] | + | + | − | − | ? | + | ? | ? |
| Barker-Collo et al [26] | + | + | − | − | + | − | + | + |
| Boyne [27] | + | ? | ? | ? | ? | + | ? | ? |
| Buhse [28] | + | + | − | − | + | + | − | + |
| Caetano[29] | + | ? | − | − | − | + | ? | ? |
| Cao[47] | + | + | − | − | + | + | ? | ? |
| Carrasquillo[30] | + | + | − | − | + | + | + | ? |
| Castellano [24] | + | + | − | − | − | + | + | + |
| Chung [48] | ? | ? | − | − | − | + | ? | ? |
| Crowley [31] | + | + | − | − | − | + | + | ? |
| Du[32] | + | + | − | − | ? | + | ? | ? |
| El-Touky [33] | ? | ? | ? | ? | ? | − | ? | ? |
| Graumlich [34] | + | + | − | − | − | + | + | + |
| Hedegaard [35] | + | + | − | − | + | + | ? | ? |
| Ho [36] | + | + | ? | ? | + | + | + | + |
| Jeong [37] | ? | ? | − | − | − | + | − | ? |
| Jia [49] | ? | + | ? | ? | ? | − | ? | ? |
| Kronish [38] | + | + | − | − | + | + | ? | ? |
| Lin [50] | + | ? | ? | ? | + | + | ? | ? |
| Marin [56] | + | ? | ? | ? | ? | + | − | − |
| Marquez-Contreras [39] | + | + | ? | ? | ? | − | ? | ? |
| Meng [23] | − | − | − | − | − | + | ? | ? |
| Meng [40] | + | + | + | + | ? | + | + | ? |
| Peng [51] | + | ? | ? | ? | ? | ? | + | ? |
| Pladevall[41] | + | ? | − | − | + | + | ? | − |
| Rinfret [42] | + | + | − | − | + | + | ? | ? |
| Samtia [54] | ? | ? | ? | ? | ? | + | ? | − |
| Schou [43] | + | + | − | − | − | + | + | − |
| Schwalm[44] | + | − | ? | ? | + | + | + | − |
| Su et al [52] | + | ? | ? | ? | ? | − | ? | ? |
| Vollmer [45] | + | ? | − | − | − | + | ? | ? |
| Volpp [46] | + | ? | − | − | + | + | − | ? |
| Xavier [55] | + | ? | − | − | − | + | + | ? |
| Xin [53] | ? | ? | − | − | + | + | ? | ? |

Key: + low risk of bias, −high risk of bias, ? unclear risk of bias.

Using the PRECIS-2 tool retrospectively posed a number of challenges. Missing data in published manuscripts is a known problem when conducting systematic reviews, and posed additional problems for this review as manuscripts often did not include the level of detail required to accurately score certain domains (online supplementary etable 6). As has been identified in previous research,[59] the most challenging domains to score were those relating to the flexibility of delivery and flexibility of participant adherence. Most studies focused on detail the content of the intervention not its delivery and adherence. To define a study as having a more pragmatic or

explanatory tendencies, we averaged domain scores to >3 or ≤3, respectively. While considered the best method, we were aware of problems which could arise as a result of combining data in this way. Combining scores could result in two or more studies receiving the same or similar scores even though individual domain scores were very different. It also meant that each domain was given equal weighting in the overall score of pragmatism. While a known limitation to the authors, no published guidance on how best to report the data could be found.

We restricted our review to articles published since 2013 to provide a contemporary update following a large Cochrane review published 2014 by Nieuwlaat *et al*.[60] Similarly to the 2014 review, our findings suggest that interventions were diverse in nature and the majority were complex, involving multiple different components. Of the number of 'successful' interventions, multifaceted interventions which included an element of education alongside regular patient contact showed the most promise, suggesting that frequent engagement with the healthcare team may trigger behavioural change or act as a reminder to undertake the behaviour. While promising, considerations need to be made as to the ability of such interventions to be upscaled and implemented. Multifaceted interventions can be expensive and therefore their cost utility needs to be explored prior to such interventions becoming embedded in clinical care pathways.

As the number of people with access to mobile technologies has increased in the past decade, particularly in the over 60 age group, interventions which rely on frequent patient contact are becoming increasingly plausible and contact via either calls or SMS provide both a pragmatic and cheap alternative to face-to-face healthcare professional contact.[61] A systematic review by Changizi and Kaveh looking into the effectiveness of mobile health in the elderly showed that it can be used effectively and is widely accepted as a source of health literacy, particularly SMS messages due to their low requirement for technological competency.[62]

Our review highlighted a number of limitations to medication adherence research conducted to date. First, the lack of gold standard method of measuring adherence makes it difficult to pool and compare outcomes. The most frequently used method within this review was self-report, particularly the tools developed by Morisky and colleagues.[57 58] These tools are quick, easy and cost-effective to administer making them ideal for use within a clinical care setting. The tools are validated and have shown moderate comparability with other indirect methods of medication adherence; however, their use has been associated with overestimation of true treatment adherence as they carry the potential for recall bias.[63]

Even among the studies using the Morisky scales methods of scoring and presenting data varied. For example, in the case of the Morisky-Green Scale, some used a Likert scoring system,[31] some classed answers of no to all questions as indicative of good adherence,[44] whereas others classed good adherence as an answer of

yes to all.[53] For the MMAS-8, studies such as Cao *et al*[47] reported mean patient scores, with higher scores representing greater adherence. In contrast, Al-Haj Mohd *et al*[25] report the proportion of people categorised as low medium or highly adherent based on a score of <6, 6–7.9 or >8, respectively. Discrepancies in reporting make it difficult to compare outcomes and therefore there is a need for standardisation. The best method to measure medication adherence across disease populations, and the best approach to reporting said results, therefore continues to be an area requiring further exploration.

We also acknowledge that our findings may be limited by our strict inclusion criteria. All papers reviewed were written in English and published since 2013 and therefore results may not represent non-English and older research. Finally the lack of quality of included studies, particularly in relation to participant blinding and reporting bias could compromise the integrity of review findings. In addition, there was some indication of publication bias for the meta-analysis of ORs (p=0.041). This suggest that some smaller studies showing a negative result might not have been published; therefore, the results of this meta-analysis need to be interpreted with some caution.

## CONCLUSION

This systematic review showed that interventions had a significant effect on improving medication adherence in populations with T2DM and/or CVD. Multifaceted interventions which included either regular patient contact or an element of education had the most significant effect. There is, however, a need to compare more standardised interventions and assess these using more uniform methods of measuring medication adherence to enable studies to be more realistically compared.

With regard to trial design, recently there has been a focus on designing trials that are pragmatic and therefore more representative of 'real life'. The findings from this review suggest that the effectiveness of interventions between pragmatic and explanatory trials was comparable, suggesting that findings can be transferred from idealised to real-word conditions. There is, however, a need for further guidance to be developed to assist researchers in characterising and scoring studies.

**Author affiliations**
[1]Diabetes Research Centre, University of Leicester, Leicester, UK
[2]Leicester Diabetes Centre, University Hospitals of Leicester NHS Trust, Leicester General Hospital, Leicester, UK
[3]Department of Oncology and Metabolism, University of Sheffield, Sheffield, UK
[4]Department of Cardiovascular Sciences, University of Leicester, Leicester, UK
[5]Department of Chemical Pathology and Metabolic Diseases, University Hospitals of Leicester NHS Trust, Leicester, UK
[6]NIHR CLAHRC East Midlands, Leicester, UK
[7]NIHR ARC East Midlands, Leicester, UK

**Acknowledgements** The authors acknowledge the financial support received from the National Institute for Health Research Collaboration for Leadership in Applied Health Research and Care—East Midlands (NIHR CLAHRC—EM), NIHR Applied

Research Collaboration (NIHR-ARC) East Midlands and the Leicester Biomedical Research Centre.

**Contributors** CF developed the study protocol, carried out the scientific literature search, screened and extracted data, undertook PRECIS-2 scoring, quality assessed studies, interpreted the results and developed the first draft of the manuscript. CG quality assessed studies, interpreted the results, undertook data checking, PRECIS-2 scoring, performed all statistical analysis and contributed to the development of the manuscript. SS assisted with the design of the review and provided intellectual content to the drafted manuscript. DK assisted with the design of the review, applied for funding and assisted with protocol development. EI screened articles for inclusion in the review. MJD provided intellectual content to the drafted manuscript. PP and PG assisted with the design of the review. KK had the concept of the study, applied for funding, assisted with protocol development and provided intellectual oversight on the development of the manuscript. KK accepts full responsibility for the conduct of the study, has access to the data and controlled the decision to publish. KK as the corresponding author attests that all listed authors meet authorship criteria and that no others meeting the criteria have been omitted.

**Funding** The authors received funding from the NIHR CLAHRC—EM and the Leicester Biomedical Research Centre.

**Disclaimer** The views expressed are those of the authors and not necessarily those of the National Health Service, the NIHR or the Department of Health.

**Competing interests** MJD reports grants from Novo Nordisk, grants from Sanofi-Aventis, grants from Lilly, grants from Boehringer Ingelheim, grants from Janssen, personal fees from Novo Nordisk, personal fees from Sanofi-Aventis, personal fees from Lilly, personal fees from Merck Sharp & Dohme, personal fees from Boehringer Ingelheim, personal fees from AstraZeneca, personal fees from Janssen, personal fees from Servier, personal fees from Mitsubishi Tanabe Pharma, personal fees from Takeda Pharmaceuticals International, outside the submitted work. KK reports personal fees from Amgen, personal fees from AstraZeneca, personal fees from Bayer, personal fees from NAPP, personal fees from Lilly, personal fees from Merck Sharp & Dohme, personal fees from Novartis, personal fees from Novo Nordisk, personal fees from Roche, personal fees from Berlin-Chemie AG/Menarini Group, personal fees from Sanofi-Aventis, personal fees from Servier, personal fees from Boehringer Ingelheim, grants from Pfizer, grants from Boehringer Ingelheim, grants from AstraZeneca, grants from Novartis, grants from Novo Nordisk, grants from Sanofi-Aventis, grants from Lilly, grants from Merck Sharp & Dohme, grants from Servier, outside the submitted work.

**Patient and public involvement** Patient and public involvement (PPI) was undertaken prior to the development of the review to obtain an understanding of patient reasons for non-adherence. A number of themes were identified and included side effects of drugs and poor explanations from healthcare professionals about prescribed drugs. Following the completion of the review, alongside the views and opinions expressed during the PPI, the study team aim to develop a toolbox and education programme to support medication adherence. A second focus group is planned comprising health professionals, with the objective of gathering their perspectives on how to promote the 'toolbox' for general utilisation in clinical practice and how best to disseminate and mainstream the toolbox of interventions.

**Patient consent for publication** Not required.

**Provenance and peer review** Not commissioned; externally peer reviewed.

**Data availability statement** Data are available on reasonable request. Data are available on reasonable request from the corresponding author.

**ORCID iD**
Kamlesh Khunti http://orcid.org/0000-0003-2343-7099

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
