## [Reviewer comments · BMJ Open]

ARTICLE DETAILS

TITLE (PROVISIONAL)	Effect of pragmatic versus explanatory interventions on medication adherence in people with cardiometabolic conditions: a systematic review and meta-analysis
AUTHORS	Fitzpatrick, Claire; Gillies, Clare; Seidu, Samuel; Kar, Debasish; Ioannidou, Ekaterini; Davies, Melanie; Patel, Prashanth; Gupta, Pankaj; Khunti, Kamlesh

VERSION 1 – REVIEW

REVIEWER	Julie Lauffenburger BWH, USA
REVIEW RETURNED	24-Jan-2020

GENERAL COMMENTS	This is a well-conducted study on an interesting topic. I have primarily minor comments on this manuscript. Minor comments: - The abstract should provide some information about how trials were classified as “explanatory” or “pragmatic” by the authors. This is not an extremely clear determination (and determined based on PRECIS criteria as described in the strengths and limitations, but not the abstract).- Abstract: What does a “standardized mean difference” mean in this context?- Background: The authors’ rationale for the need for this study stems from the PRECIS-2 criteria being applied during the initial review as opposed to retrospectively, but it’s not clear why this really adds meaningfully.- There was some evidence of publication bias in some of the outcomes – the authors should speak to whether this lowers the strength of their conclusions. Discretionary comments: - It is not clear in the objective what “relatable” means in terms of real-world clinical practice (abstract, line 11).- Page 8, Line 12: The sentence beginning with “In one study” is out of place in the paragraph.- Page 8, Line 20-22: The authors should consider conducting subgroup analyses not necessarily by self-reported adherence vs. objective but also by administrative claims vs. other types of objective data (e.g., electronic drug monitors).
---

REVIEWER	Katerina Kassavou University of Cambridge, UK.
REVIEW RETURNED	26-Jan-2020

GENERAL COMMENTS	This paper describes a review of evidence and aims to respond to
--

	an important question: whether the design of a trial impacts on the outcomes. Although timely, there are some methodological and major conceptual limitations in this research. Currently, it reads as if it is a subgroup analysis of a non-clearly defined area of research. Researchers have used the pre-developed PRECIS-2 tool to categorise studies based on their design, which cannot per se justify a unique contribution to the knowledge. The paper could be improved if more information is included on how the study design impacts on the trial outcomes. Additionally, the coding of the intervention components has not been conducted based on existing and widely used tools, which decreases the validity of the results and the overall quality of this research. Comments.  1. The authors could include more information about how the design of the trials (e.g. pragmatic versus explanatory) could influence changes in medication adherence. There is key literature to justify authors claims about whether and how study design could bias outcomes in behaviour change and in medication adherence trials (e.g. research in methodology by Jim McCambridge), which is currently missing from the introduction. 2. A conceptual definition of 'pragmatism' could have shade light into the context of this area of research. Probably in the discussion. 3. The categorisation and definition of the interventions is not justified. The definition of intervention components (supplementary eTable 2) describes a combination of some complex forms of delivery mode and intervention content. Could not authors use an existing tool to categorise intervention components? 4. It is not clear how the quantitative data syntheses were conducted (e.g. meta-regression and sub-group analysis). What was the rationale to undertake a meta-regression or a sub-group analyses on these variables? it also seems that the primary research question (i.e. explanatory vs. pragmatic) was investigated using sub-group analyses, instead of meta-regression. Could authors justify the reason for this? Minor comments  1. The authors report that 'change in adherence for each study arm was calculated'. Could they provide more information about how they measure 'changes in medication adherence'? what primary outcome data was included in the analysis? (e.g. adjusted or unadjusted data). 2. Please describe in more details how the authors decided on what outcome data to include in meta-analysis e.g. were any studies that included more than two outcome measurements? how did they decide on which outcome measure to include in the analyses? 3. Risk of bias. The authors seem that they have used the Cochrane risk of bias tool. Does this differ to the revised Cochrane risk of bias tool for randomised controlled trials? 4. Table 1 might be useful if it was included in one page.
--	--

VERSION 1 – AUTHOR RESPONSE

Response to reviewer 1 comments

1. The abstract should provide some information about how trials were classified as “explanatory” or “pragmatic” by the authors. This is not an extremely clear determination (and determined based on

PRECIS criteria as described in the strengths and limitations, but not the abstract)

We thank the reviewer for this helpful comment. We have now explained the terms in the introductory sentence in the abstract.

The following was changed from:

To synthesise findings from randomised controlled trials (RCTs) of interventions aimed at increasing medication adherence in individuals with type 2 diabetes (T2DM) and/or cardiovascular disease (CVD). And to compare the effect of explanatory versus pragmatic trial designs to explore whether outcomes are relatable to real world clinical practice.

To:

To synthesise findings from randomised controlled trials (RCTs) of interventions aimed at increasing medication adherence in individuals with type 2 diabetes (T2DM) and/or cardiovascular disease (CVD). And, in a novel approach, to compare the intervention effect of studies which were categorised as being more pragmatic or more explanatory using the Pragmatic-Explanatory Continuum Indicator Summary-2 (PRECIS-2) tool, to identify whether study design affects outcomes. As explanatory trials are typically held under controlled conditions, findings from such trials may not be relatable to real world clinical practice. In comparison, pragmatic trials are designed to replicate real world conditions and therefore findings are more likely to represent those found if the intervention were to be implemented in routine care.

2. Abstract: What does a “standardized mean difference” mean in this context?

The following sentence has been added to the Methods – data synthesis section of the article. “The standardised mean difference was calculated as the mean change in medication adherence in the intervention group minus mean change in the control group, divided by the pooled standard deviation, using Cohen’s method.”

Reference

Cohen J. Statistical power analysis for the behavioral sciences: Routledge; 2013.

3. Background: The authors’ rationale for the need for this study stems from the PRECIS-2 criteria being applied during the initial review as opposed to retrospectively, but it’s not clear why this really adds meaningfully.

We thank the reviewer once again for this point. We have now provided a clarification on this point in the introduction.

The following paragraph has been added to the last paragraph of the introduction.

Whilst retrospective application provides an interesting comparison, whereby the overall intervention effects can be compared to explanatory and/or pragmatic intervention effects, pre or post use of PRECIS-2, it could lead to an initial misinterpretation of findings. For example, a review containing a high number of explanatory trials may be much less applicable to routine clinical practice than one containing a greater number of pragmatic trials. Identification and comparison during the trial design stage would allow researchers to more easily identify how applicable findings would be to real world clinical practice, rather than making an assumption based on a generalised outcome. In addition, this approach removes any risk of bias as the analysts are unaware of the results of the study results.

4. There was some evidence of publication bias in some of the outcomes – the authors should speak to whether this lowers the strength of their conclusions.

The following sentence was added to the last paragraph of the discussion.

In addition, there was some indication of publication bias for the meta-analysis of odds ratios ($p=0.041$). This suggests that some smaller studies showing a negative result might not have been published, therefore the results of this meta-analysis need to be interpreted with some caution.

5. It is not clear in the objective what “relatable” means in terms of real-world clinical practice (abstract, line 11).

The following was changed from:

To synthesise findings from randomised controlled trials (RCTs) of interventions aimed at increasing medication adherence in individuals with type 2 diabetes (T2DM) and/or cardiovascular disease (CVD). And to compare the effect of explanatory versus pragmatic trial designs to explore whether outcomes are relatable to real world clinical practice.

To:

And, in a novel approach, to compare the intervention effect of studies which were categorised as being more pragmatic or more explanatory using the Pragmatic-Explanatory Continuum Indicator Summary-2 (PRECIS-2) tool in a meta-analysis to identify whether study design affects outcomes. As explanatory trials are typically held under controlled conditions, findings from such trials may not be relatable to real world clinical practice. In comparison, pragmatic trials are designed to replicate real world conditions and therefore findings are more likely to represent those found if the intervention were to be implemented in routine care.

6. Page 8, Line 12: The sentence beginning with “In one study” is out of place in the paragraph

The sentence has now been moved up to a previous paragraph where the methods for the odds ratio calculations are discussed.

7. Page 8, Line 20-22: The authors should consider conducting subgroup analyses not necessarily by self-reported adherence vs. objective but also by administrative claims vs. other types of objective data (e.g., electronic drug monitors)

We thank the reviewer for this suggestion however, of the objective measures, there were ten studies that looked at pharmacy data, two that looked at pill count and one that used electronic pill bottle opening as a measure of adherence. Therefore we feel there are too few studies in two of these categories to carry out further sub-group analyses.

Response to reviewer 2 comments

1. The authors could include more information about how the design of the trials (e.g. pragmatic versus explanatory) could influence changes in medication adherence. There is key literature to justify authors claims about whether and how study design could bias outcomes in behaviour change and in medication adherence trials (e.g. research in methodology by Jim McCambridge), which is currently missing from the introduction.

We wish to thank the reviewer for this important suggestion. We have now clarified this in the introduction and cited the suggested reference.

Whilst randomised controlled trials (RCTs) are widely accepted as a rigorous way of exploring the impact of interventions on specific health behaviour change outcomes, it has been identified that

numerous components within a trials design can lead to biased interpretations of intervention effects.(9) One such bias may be the controlled nature of these trials and the impact they impose upon the cooperation of their participants, and may prelude an action which does not necessarily represent what may occur in routine clinical practice. In pragmatic trials, the intervention is less strict and mimics usual practice as much as possible, thus lessening the unexpected reactions from the patients which lead to the biases.

2. A conceptual definition of 'pragmatism' could have shade light into the context of this area of research. Probably in the discussion.

The following sentence has been added to the introduction to explain the importance of why we are wanting to compare pragmatic versus explanatory intervention effects.

Described by Schwartz & Lellouch (1967),(9) explanatory trial are those which confirm a physiological or clinical hypothesis, in contrast, pragmatic trails are those which inform clinical or policy decisions by evidencing the effect that adoption would have on routine care. As treatment effects of explanatory trials may be larger than those observed in pragmatic trials, traditional meta-analytic approaches may not account for this heterogeneity resulting in biased estimated treatment effects.

3. The categorisation and definition of the interventions is not justified. The definition of intervention components (supplementary eTable 2) describes a combination of some complex forms of delivery mode and intervention content. Could not authors use an existing tool to categorise intervention components?

We thank the reviewer once again for this suggestion. Upon searching we were unable to find an existing tool to categorise such intervention components. As such these were devised using previous adherence research for guidance. A footnote has been added to the table referencing the guidance documents. We used the commonly occurring themes in the studies used for the analysis to identify the categories; Tele monitoring, Behavioural/educational, Collaborative care, Simplification of drug regimen, Intensified patient care, Personalised drug dispensing, and multi-component interventions comprising various combinations of the listed components.

4. It is not clear how the quantitative data syntheses were conducted (e.g. meta-regression and sub-group analysis). What was the rationale to undertake a meta-regression or a sub-group analyses on these variables? it also seems that the primary research question (i.e. explanatory vs. pragmatic) was investigated using sub-group analyses, instead of meta-regression. Could authors justify the reason for this?

We had used meta-regression models to assess association between the study effect size of medication adherence and a number of variables. For the sub-group analyses, as well as fitting a meta-regression model to assess the difference between the two sub-groups, we also ran the meta-analyses for each sub-group. This allowed the pooled estimate for each sub-group to be calculated, and for a visual comparison between sub-groups on a forest plot. The p-values from the meta-regression models comparing subgroups are reported in supplementary table eTable 5. To clarify this in the text, we have now changed the text (new text in red):

"We fitted meta-regression models assessing study length, whether adherence was a primary or secondary outcome, mean age, percent male, disease (T2DM or CVD), and PRECIS-2 score, to explore the impact of study heterogeneity on the intervention effect. Three sub-group analyses were also carried out, whereby as well as fitting a meta-regression model to compare the statistical difference between groups, separate meta-analyses were also run for each subgroup to enable the pooled estimate for each sub-group to be calculated, and hence a more explicit comparison to be

made. The first compared intervention effects of studies that included a self-reported measure of adherence, to those with an objective adherence measure. The second compared pragmatic with explanatory studies (pragmatic studies were those with an average PRECIS-2 score >3 and explanatory a score of 3 or less) and the third compared outcomes of interventions which were identified as multi-faceted against those with a singular intervention component.”

Additional comments

1. The authors report that ‘change in adherence for each study arm was calculated’. Could they provide more information about how they measure ‘changes in medication adherence’? what primary outcome data was included in the analysis? (e.g. adjusted or unadjusted data).

We have clarified how adherence was calculated in the text, as well as that adjusted ORs were chosen over unadjusted (new text in red):

Study data were reported as means and medians for continuous data and as proportions for categorical data. Twenty two studies defined a cut point to determine adherence. Twenty used a binary outcome defining individuals as adherent or non-adherent, however two categorised individuals based on a pre-specified level of low, medium, and high. Of those reporting levels of low, medium or high adherence, as has been done in previous studies,(18, 19) we combined medium and high levels and separated low level to form a binary outcome of adherence or non-adherence respectively, therefore enabling odds ratio of adherence to be calculated. In one study (Boyne, 2014), all participants in the intervention arm were adherent at follow-up, so a continuity correction of 0.5 was added to allow an odds ratio to be calculated. Where an odds ratio for medication adherence was reported in study results, this was used for the meta-analyses, rather than calculating an estimated odds ratio from the raw numbers. Where adjusted odds ratios were reported, the odds ratio adjusted for the most covariates was used in the meta-analyses.

A further six studies, plus one which was included in the previous analysis, reported adherence using a continuous scale which provided an overall group indication of adherence. For these studies, change in adherence for each study arm was calculated. As scores differed, we calculated standardised mean differences (SMD) to combine study estimates in a meta-analysis. The standardised mean difference was calculated as the mean change in medication adherence in the intervention group – mean change in the control group, divided by the pooled standard deviation, using Cohen’s method.

2. Please describe in more details how the authors decided on what outcome data to include in meta-analysis e.g. were any studies that included more than two outcome measurements? how did they decide on which outcome measure to include in the analyses?

Only one study reported both a mean change and an odds ratio for medication adherence, and was included in both meta-analyses. Where a study reported two outcomes on the same scale we chose the outcome that was given priority in the results- either because it was listed as the primary outcome, or because it was listed first when the outcomes were described.

3. Risk of bias. The authors seem that they have used the Cochrane risk of bias tool. Does this differ to the revised Cochrane risk of bias tool for randomised controlled trials?

Risk of bias has been re-assessed using the revised Cochrane risk of bias tool and the reference has been updated in the text and blinding or participants, personnel and outcome assessors in now listed as three separate columns rather than two in table 2. Utilisation of the revised tool did not have any impact upon the overall assessment of bias for any of the included articles.

4. Table 1 might be useful if it was included in one page.

This has been edited so that the table now fits on one page

VERSION 2 – REVIEW

REVIEWER	Julie Lauffenburger Brigham and Women's, USA
REVIEW RETURNED	27-Mar-2020
GENERAL COMMENTS	In my view, the authors have been sufficiently responsive to prior reviewer comments. I have some minor comments around presentation, outlined below: - Figures 2 and 3: The headings "pragmatic" and "explanatory" are themselves not explanatory without the context of the paper. Please add descriptors to how these categories were determined so that this figure can stand alone. Also, $p=0.000$ should likely be $p<0.001$ instead.- It might also help to have a table in the main paper with simplified characteristics of the study. Without going to the supplement, it is not clear which papers were part of the systematic review.- Conclusions in abstract and conclusions in the manuscript are not exactly aligned -- recommend reconciling.

VERSION 2 – AUTHOR RESPONSE

Response to reviewer's comments

1. Figures 2 and 3: The headings "pragmatic" and "explanatory" are themselves not explanatory without the context of the paper. Please add descriptors to how these categories were determined so that this figure can stand alone. Also, $p=0.000$ should likely be $p<0.001$ instead.

We thank the reviewer for this helpful comment. We have now defined the terms and changed $p=0.000$ to $p<0.001$.

2. It might also help to have a table in the main paper with simplified characteristics of the study. Without going to the supplement, it is not clear which papers were part of the systematic review.

We thank the reviewer for this suggested and in response we have added a summary characteristics table to the manuscript (Table 1, page 10).

3. Conclusions in abstract and conclusions in the manuscript are not exactly aligned -- recommend reconciling.

We once again thank the reviewer for this comment and have amended the conclusion of the article to bring the abstract and article inclusion inline. The article conclusion has been changed from: This systematic review showed that interventions worked effectively to improve medication adherence in populations with T2DM and/or CVD. Multi-faceted interventions had the most significant effect. There is however a need to compare more standardised interventions and assess these using more uniform methods of measuring medication adherence to enable studies to be more realistically compared.

Recently there has been a focus on designing trials that are pragmatic and therefore more representative of 'real life' and the findings from this review suggest that these interventions are

equally effective. There is however a need for further guidance to be developed to assist researchers in characterising and scoring studies.

To:

This systematic review showed that interventions had a significant effect on improving medication adherence in populations with T2DM and/or CVD. Multi-faceted interventions which included either regular patient contact or an element of education had the most significant effect. There is however a need to compare more standardised interventions and assess these using more uniform methods of measuring medication adherence to enable studies to be more realistically compared.

With regards to trial design, recently there has been a focus on designing trials that are pragmatic and therefore more representative of 'real life'. The findings from this review suggest that the effectiveness of interventions between pragmatic and explanatory trials was comparable, suggesting that findings can be transferred from idealised to real word conditions. There is however a need for further guidance to be developed to assist researchers in characterising and scoring studies.

Additional comments from authors

We apologise for our initial oversight however whilst revising the manuscript we noticed some formatting errors with regard to references in the Included Studies Characteristics Table in the supplementary materials document. References have therefore been updated and a new version of the supplementary materials document has been uploaded.